# Generating Images with Perceptual Similarity Metrics based on Deep Networks

**Alexey Dosovitskiy and Thomas Brox**
University of Freiburg
{dosovits, brox}@cs.uni-freiburg.de

## Abstract

We propose a class of loss functions, which we call deep perceptual similarity metrics (DeePSiM), allowing to generate sharp high resolution images from compressed abstract representations. Instead of computing distances in the image space, we compute distances between image features extracted by deep neural networks. This metric reflects perceptual similarity of images much better and, thus, leads to better results. We demonstrate two examples of use cases of the proposed loss: (1) networks that invert the AlexNet convolutional network; (2) a modified version of a variational autoencoder that generates realistic high-resolution random images.

## 1 Introduction

Recently there has been a surge of interest in training neural networks to generate images. These are being used for a wide variety of applications: generative models, analysis of learned representations, learning of 3D representations, future prediction in videos. Nevertheless, there is little work on studying loss functions which are appropriate for the image generation task. The widely used squared Euclidean (SE) distance between images often yields blurry results; see Fig. 1 (b). This is especially the case when there is inherent uncertainty in the prediction. For example, suppose we aim to reconstruct an image from its feature representation. The precise location of all details is not preserved in the features. A loss in image space leads to averaging all likely locations of details, hence the reconstruction looks blurry.

However, exact locations of all fine details are not important for perceptual similarity of images. What is important is the distribution of these details. Our main insight is that invariance to irrelevant transformations and sensitivity to local image statistics can be achieved by measuring distances in a suitable feature space. In fact, convolutional networks provide a feature representation with desirable properties. They are invariant to small, smooth deformations but sensitive to perceptually important image properties, like salient edges and textures.

Using a distance in feature space alone does not yet yield a good loss function; see Fig. 1 (d). Since feature representations are typically contractive, feature similarity does not automatically mean image similarity. In practice this leads to high-frequency artifacts (Fig. 1 (d)). To force the network generate realistic images, we introduce a natural image prior based on adversarial training, as proposed by Goodfellow et al. [1] [1]. We train a discriminator network to distinguish the output of the generator from real images based on local image statistics. The objective of the generator is to trick the discriminator, that is, to generate images that the discriminator cannot distinguish from real ones. A combination of similarity in an appropriate feature space with adversarial training yields the best results; see Fig. 1 (e). Results produced with adversarial loss alone (Fig. 1 (c)) are clearly inferior to those of our approach, so the feature space loss is crucial.

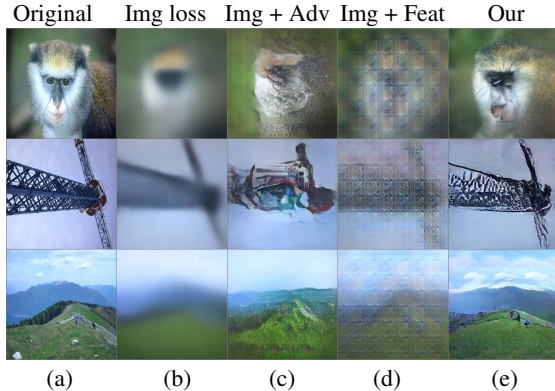

Original    Img loss    Img + Adv    Img + Feat    Our

(a)     (b)     (c)     (d)     (e)

Figure 1: Reconstructions from AlexNet FC6 with different components of the loss.

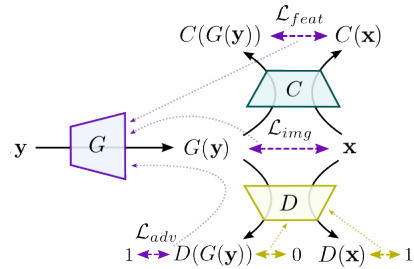

Figure 2: Schematic of our model. Black solid lines denote the forward pass. Dashed lines with arrows on both ends are the losses. Thin dashed lines denote the flow of gradients.

The new loss function is well suited for generating images from highly compressed representations. We demonstrate this in two applications: inversion of the AlexNet convolutional network and a generative model based on a variational autoencoder. Reconstructions obtained with our method from high-level activations of AlexNet are significantly better than with existing approaches. They reveal that even the predicted class probabilities contain rich texture, color, and position information. As an example of a true generative model, we show that a variational autoencoder trained with the new loss produces sharp and realistic high-resolution $227 \times 227$ pixel images.

## 2 Related work

There is a long history of neural network based models for image generation. A prominent class of probabilistic models of images are restricted Boltzmann machines [3] and their deep variants [4, 5]. Autoencoders [6] have been widely used for unsupervised learning and generative modeling, too. Recently, stochastic neural networks [7] have become popular, and deterministic networks are being used for image generation tasks [8]. In all these models, loss is measured in the image space. By combining convolutions and un-pooling (upsampling) layers [5, 1, 8] these models can be applied to large images.

There is a large body of work on assessing the perceptual similarity of images. Some prominent examples are the visible differences predictor [9], the spatio-temporal model for moving picture quality assessment [10], and the perceptual distortion metric of Winkler [11]. The most popular perceptual image similarity metric is the structural similarity metric (SSIM) [12], which compares the local statistics of image patches. We are not aware of any work making use of similarity metrics for machine learning, except a recent pre-print of Ridgeway et al. [13]. They train autoencoders by directly maximizing the SSIM similarity of images. This resembles in spirit what we do, but technically is very different. Because of its shallow and local nature, SSIM does not have invariance properties needed for the tasks we are solving in this paper.

Generative adversarial networks (GANs) have been proposed by Goodfellow et al. [1]. In theory, this training procedure can lead to a generator that perfectly models the data distribution. Practically, training GANs is difficult and often leads to oscillatory behavior, divergence, or modeling only part of the data distribution. Recently, several modifications have been proposed that make GAN training more stable. Denton et al. [14] employ a multi-scale approach, gradually generating higher resolution images. Radford et al. [15] make use of an upconvolutional architecture and batch normalization.

GANs can be trained conditionally by feeding the conditioning variable to both the discriminator and the generator [16]. Usually this conditioning variable is a one-hot encoding of the object class in the input image. Such GANs learn to generate images of objects from a given class. Recently Mathieu et al. [17] used GANs for predicting future frames in videos by conditioning on previous frames. Our approach looks similar to a conditional GAN. However, in a GAN there is no loss directly comparing the generated image to some ground truth. As Fig. 1 shows, the feature loss introduced in the present paper is essential to train on complicated tasks we are interested in.

Several concurrent works [18–20] share the general idea — to measure the similarity not in the image space but rather in a feature space. These differ from our work both in the details of the method and in the applications. Larsen et al. [18] only run relatively small-scale experiments on images of faces, and they measure the similarity between features extracted from the discriminator, while we study different "comparators" (in fact, we also experimented with features from the disciminator and were not able to get satisfactory results on our applications with those). Lamb et al. [19] and Johnson et al. [20] use features from different layers, including the lower ones, to measure image similarity, and therefore do not need the adversarial loss. While this approach may be suitable for tasks which allow for nearly perfect solutions (e.g. super-resolution with low magnification), it is not applicable to more complicated problems such as extreme super-resolution or inversion of highly invariant feature representations.

## 3 Model

Suppose we are given a supervised image generation task and a training set of input-target pairs $\{\mathbf{y}_i, \mathbf{x}_i\}$, consisting of high-level image representations $\mathbf{y}_i \in \mathbb{R}^I$ and images $\mathbf{x}_i \in \mathbb{R}^{W \times H \times C}$ .

The aim is to learn the parameters $\theta$ of a differentiable generator function $G_\theta(\cdot) \colon \mathbb{R}^I \to \mathbb{R}^{W \times H \times C}$ which optimally approximates the input-target dependency according to a loss function $\mathcal{L}(G_\theta(\mathbf{y}), \mathbf{x})$. Typical choices are squared Euclidean (SE) loss $\mathcal{L}_2(G_\theta(\mathbf{y}), \mathbf{x}) = ||G_\theta(\mathbf{y}) - \mathbf{x}||_2^2$ or $\ell_1$ loss $\mathcal{L}_1(G_\theta(\mathbf{y}), \mathbf{x}) = ||G_\theta(\mathbf{y}) - \mathbf{x}||_1$, but these lead to blurred results in many image generation tasks.

We propose a new class of losses, which we call deep perceptual similarity metrics (DeePSiM ). These go beyond simple distances in image space and can capture complex and perceptually important properties of images. These losses are weighted sums of three terms: feature loss $\mathcal{L}_{feat}$, adversarial loss $\mathcal{L}_{adv}$, and image space loss $\mathcal{L}_{img}$:

$$\mathcal{L} = \lambda_{feat}\,\mathcal{L}_{feat} + \lambda_{adv}\,\mathcal{L}_{adv} + \lambda_{img}\,\mathcal{L}_{img}. \tag{1}$$

They correspond to a network architecture, an overview of which is shown in Fig. 2 . The architecture consists of three convolutional networks: the generator $G_\theta$ that implements the generator function, the discriminator $D_\varphi$ that discriminates generated images from natural images, and the comparator $C$ that computes features used to compare the images.

**Loss in feature space.** Given a differentiable comparator $C \colon \mathbb{R}^{W \times H \times C} \to \mathbb{R}^F$, we define

$$\mathcal{L}_{feat} = \sum_i ||C(G_\theta(\mathbf{y}_i)) - C(\mathbf{x}_i)||_2^2. \tag{2}$$

$C$ may be fixed or may be trained; for example, it can be a part of the generator or the discriminator.

$\mathcal{L}_{feat}$ alone does not provide a good loss for training. Optimizing just for similarity in a high-level feature space typically leads to high-frequency artifacts [21]. This is because for each natural image there are many non-natural images mapped to the same feature vector [2] . Therefore, a natural image prior is necessary to constrain the generated images to the manifold of natural images.

**Adversarial loss.** Instead of manually designing a prior, as in Mahendran and Vedaldi [21], we learn it with an approach similar to Generative Adversarial Networks (GANs) of Goodfellow et al. [1] . Namely, we introduce a discriminator $D_\varphi$ which aims to discriminate the generated images from real ones, and which is trained concurrently with the generator $G_\theta$. The generator is trained to "trick" the discriminator network into classifying the generated images as real. Formally, the parameters $\varphi$ of the discriminator are trained by minimizing

$$\mathcal{L}_{discr} = -\sum_i \log(D_\varphi(\mathbf{x}_i)) + \log(1 - D_\varphi(G_\theta(\mathbf{y}_i))), \tag{3}$$

and the generator is trained to minimize

$$\mathcal{L}_{adv} = -\sum_i \log D_\varphi(G_\theta(\mathbf{y}_i)). \tag{4}$$

**Loss in image space.** Adversarial training is unstable and sensitive to hyperparameter values. To suppress oscillatory behavior and provide strong gradients during training, we add to our loss function a small squared error term:

$$\mathcal{L}_{img} = \sum_i ||G_\theta(\mathbf{y}_i) - \mathbf{x}_i||_2^2. \tag{5}$$

We found that this term makes hyperparameter tuning significantly easier, although it is not strictly necessary for the approach to work.

### 3.1 Architectures

**Generators.** All our generators make use of up-convolutional ('deconvolutional') layers [8]. An up-convolutional layer can be seen as up-sampling and a subsequent convolution. We always up-sample by a factor of 2 with 'bed of nails' upsampling. A basic generator architecture is shown in Table 1.

In all networks we use leaky ReLU nonlinearities, that is, $LReLU(x) = \max(x, 0) + \alpha \min(x, 0)$. We used $\alpha = 0.3$ in our experiments. All generators have linear output layers.

**Comparators.** We experimented with three comparators:

1. AlexNet [22] is a network with 5 convolutional and 2 fully connected layers trained on image classification. More precisely, in all experiments we used a variant of AlexNet called CaffeNet [23].

2. The network of Wang and Gupta [24] has the same architecture as CaffeNet, but is trained without supervision. The network is trained to map frames of one video clip close to each other in the feature space and map frames from different videos far apart. We refer to this network as VideoNet.

3. AlexNet with random weights.

We found using CONV5 features for comparison leads to best results in most cases. We used these features unless specified otherwise.

**Discriminator.** In our setup the job of the discriminator is to analyze the local statistics of images. Therefore, after five convolutional layers with occasional stride we perform global average pooling. The result is processed by two fully connected layers, followed by a 2-way softmax. We perform 50% dropout after the global average pooling layer and the first fully connected layer. The exact architecture of the discriminator is shown in the supplementary material.

### 3.2 Training details

Coefficients for adversarial and image loss were respectively $\lambda_{adv} = 100$, $\lambda_{img} = 2 \cdot 10^{-6}$. The feature loss coefficient $\lambda_{feat}$ depended on the comparator being used. It was set to $0.01$ for the AlexNet CONV5 comparator, which we used in most experiments. Note that a high coefficient in front of the adversarial loss does not mean that this loss dominates the error function; it simply compensates for the fact that both image and feature loss include summation over many spatial locations. We modified the *caffe* [23] framework to train the networks. For optimization we used Adam [25] with momentum $\beta_1 = 0.9$, $\beta_2 = 0.999$ and initial learning rate $0.0002$. To prevent the discriminator from overfitting during adversarial training we temporarily stopped updating it if the ratio of $\mathcal{L}_{discr}$ and $\mathcal{L}_{adv}$ was below a certain threshold ($0.1$ in our experiments). We used batch size 64 in all experiments. The networks were trained for $500,000$-$1,000,000$ mini-batch iterations.

## 4 Experiments

### 4.1 Inverting AlexNet

As a main application, we trained networks to reconstruct images from their features extracted by AlexNet. This is interesting for a number of reasons. First and most straightforward, this shows which information is preserved in the representation. Second, reconstruction from artificial networks can be seen as test-ground for reconstruction from real neural networks. Applying the proposed method to real brain recordings is a very exciting potential extension of our work. Third, it is interesting to see that in contrast with the standard scheme "generative pretraining for a discriminative task", we show that "discriminative pre-training for a generative task" can be fruitful. Lastly, we indirectly show that our loss can be useful for unsupervised learning with generative models. Our version of

| Type | fc | fc | fc | reshape | uconv | conv | uconv | conv | uconv | conv | uconv | uconv | uconv |
|---|---|---|---|---|---|---|---|---|---|---|---|---|---|
| InSize | – | – | – | 1 | 4 | 8 | 8 | 16 | 16 | 32 | 32 | 64 | 128 |
| OutCh | 4096 | 4096 | 4096 | 256 | 256 | 512 | 256 | 256 | 128 | 128 | 64 | 32 | 3 |
| Kernel | – | – | – | – | 4 | 3 | 4 | 3 | 4 | 3 | 4 | 4 | 4 |
| Stride | – | – | – | – | ↑2 | 1 | ↑2 | 1 | ↑2 | 1 | ↑2 | ↑2 | ↑2 |

Table 1: Generator architecture for inverting layer FC6 of AlexNet.

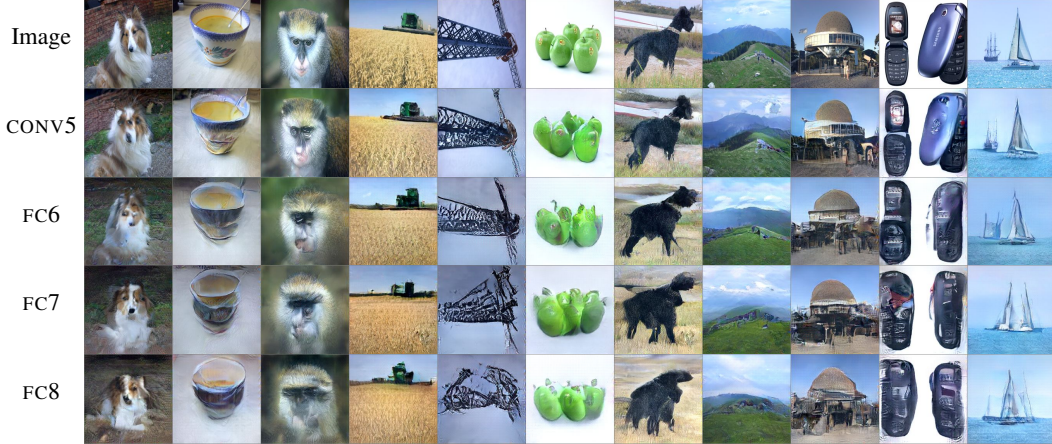

Figure 3: Representative reconstructions from higher layers of AlexNet. General characteristics of images are preserved very well. In some cases (simple objects, landscapes) reconstructions are nearly perfect even from FC8. In the leftmost column the network generates dog images from FC7 and FC8.

reconstruction error allows us to reconstruct from very abstract features. Thus, in the context of unsupervised learning, it would not be in conflict with learning such features.

We describe how our method relates to existing work on feature inversion. Suppose we are given a feature representation $\Phi$, which we aim to invert, and an image $\mathbf{I}$. There are two inverse mappings: $\Phi_R^{-1}$ such that $\Phi(\Phi_R^{-1}(\phi)) \approx \phi$, and $\Phi_L^{-1}$ such that $\Phi_L^{-1}(\Phi(\mathbf{I})) \approx \mathbf{I}$. Recently two approaches to inversion have been proposed, which correspond to these two variants of the inverse.

Mahendran and Vedaldi [21] apply gradient-based optimization to find an image $\widetilde{\mathbf{I}}$ which minimizes

$$||\Phi(\mathbf{I}) - \Phi(\widetilde{\mathbf{I}})||_2^2 + P(\widetilde{\mathbf{I}}), \tag{6}$$

where $P$ is a simple natural image prior, such as the total variation (TV) regularizer. This method produces images which are roughly natural and have features similar to the input features, corresponding to $\Phi_R^{-1}$. However, due to the simplistic prior, reconstructions from fully connected layers of AlexNet do not look much like natural images (Fig. 4 bottom row).

Dosovitskiy and Brox [26] train up-convolutional networks on a large training set of natural images to perform the inversion task. They use squared Euclidean distance in the image space as loss function, which leads to approximating $\Phi_L^{-1}$. The networks learn to reconstruct the color and rough positions of objects, but produce over-smoothed results because they average all potential reconstructions (Fig. 4 middle row).

Our method combines the best of both worlds, as shown in the top row of Fig. 4. The loss in the feature space helps preserve perceptually important image features. Adversarial training keeps reconstructions realistic.

**Technical details.** The generator in this setup takes the features $\Phi(\mathbf{I})$ extracted by AlexNet and generates the image $\mathbf{I}$ from them, that is, $\mathbf{y} = \Phi(\mathbf{I})$. In general we followed Dosovitskiy and Brox [26] in designing the generators. The only modification is that we inserted more convolutional layers, giving the network more capacity. We reconstruct from outputs of layers CONV5 –FC8. In each layer we also include processing steps following the layer, that is, pooling and non-linearities. For example, CONV5 means pooled features (pool5), and FC6 means rectified values (relu6).

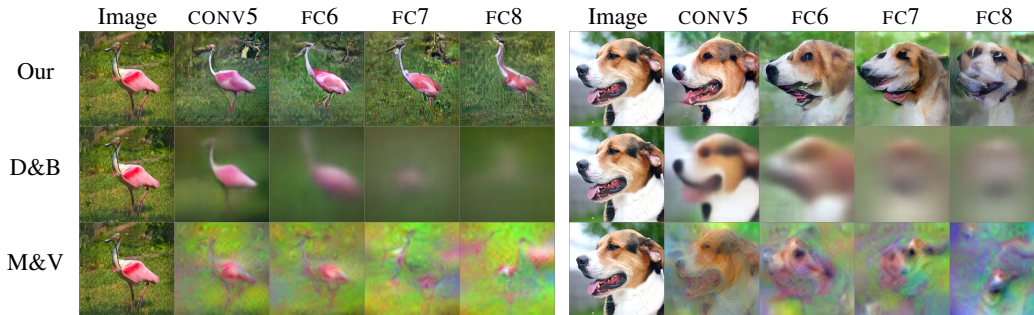

Figure 4: AlexNet inversion: comparison with Dosovitskiy and Brox [26] and Mahendran and Vedaldi [21] . Our results are significantly better, even our failure cases (second image).

The generator used for inverting FC6 is shown in Table 1 . Architectures for other layers are similar, except that for reconstruction from CONV5, fully connected layers are replaced by convolutional ones. We trained on $227 \times 227$ pixel crops of images from the ILSVRC-2012 training set and evaluated on the ILSVRC-2012 validation set.

**Ablation study.** We tested if all components of the loss are necessary. Results with some of these components removed are shown in Fig. 1 . Clearly the full model performs best. Training just with loss in the image space leads to averaging all potential reconstructions, resulting in over-smoothed images. One might imagine that adversarial training makes images sharp. This indeed happens, but the resulting reconstructions do not correspond to actual objects originally contained in the image. The reason is that any "natural-looking" image which roughly fits the blurry prediction minimizes this loss. Without the adversarial loss, predictions look very noisy because nothing enforces the natural image prior. Results without the image space loss are similar to the full model (see supplementary material), but training was more sensitive to the choice of hyperparameters.

**Inversion results.** Representative reconstructions from higher layers of AlexNet are shown in Fig. 3 . Reconstructions from CONV5 are nearly perfect, combining the natural colors and sharpness of details. Reconstructions from fully connected layers are still strikingly good, preserving the main features of images, colors, and positions of large objects. More results are shown in the supplementary material.

For quantitative evaluation we compute the normalized Euclidean error $||a - b||_2/N$. The normalization coefficient $N$ is the average of Euclidean distances between all pairs of different samples from the test set. Therefore, the error of $100\%$ means that the algorithm performs the same as randomly drawing a sample from the test set. Error in image space and in feature space (that is, the distance between the features of the image and the reconstruction) are shown in Table 2 . We report all numbers for our best approach, but only some of them for the variants, because of limited computational resources.

The method of Mahendran&Vedaldi performs well in feature space, but not in image space, the method of Dosovitskiy&Brox — vice versa. The presented approach is fairly good on both metrics. This is further supported by iterative image re-encoding results shown in Fig. 5 . To generate these, we compute the features of an image, apply our "inverse" network to those, compute the features of the resulting reconstruction, apply the "inverse" net again, and iterate this procedure. The reconstructions start to change significantly only after $4 - 8$ iterations of this process.

**Nearest neighbors** Does the network simply memorize the training set? For several validation images we show nearest neighbors (NNs) in the training set, based on distances in different feature spaces (see supplementary material). Two main conclusions are: 1) NNs in feature spaces are much more meaningful than in the image space, and 2) The network does more than just retrieving the NNs.

**Interpolation.** We can morph images into each other by linearly interpolating between their features and generating the corresponding images. Fig. 7 shows that objects shown in the images smoothly warp into each other. This capability comes "for free" with our generator networks, but in fact it is very non-trivial, and to the best of our knowledge has not been previously demonstrated to this extent on general natural images. More examples are shown in the supplementary material.

|  | CONV5 | FC6 | FC7 | FC8 |
|---|---|---|---|---|
| M & V [21] | 71/19 | 80/19 | 82/16 | 84/09 |
| D & B [26] | 35/− | 51/− | 56/− | 58/− |
| Our image loss | −/− | 46/79 | −/− | −/− |
| AlexNet CONV5 | 43/37 | 55/48 | 61/45 | 63/29 |
| VideoNet CONV5 | −/− | 51/57 | −/− | −/− |

Table 2: Normalized inversion error (in %) when reconstructing from different layers of AlexNet with different methods. First in each pair – error in the image space, second – in the feature space.



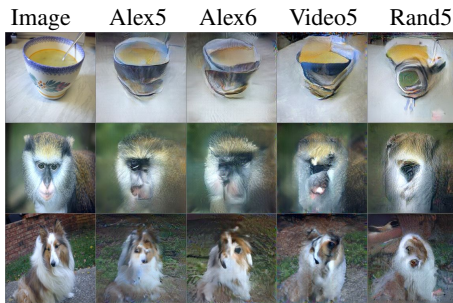

Figure 5: Iteratively re-encoding images with AlexNet and reconstructing. Iteration number shown on the left.

Figure 6: Reconstructions from FC6 with different comparators. The number indicates the layer from which features were taken.

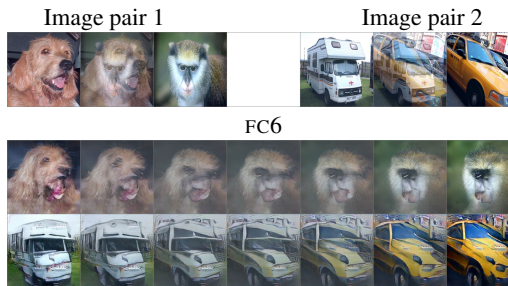

Figure 7: Interpolation between images by interpolating between their FC6 features.

**Different comparators.** The AlexNet network we used above as comparator has been trained on a huge labeled dataset. Is this supervision really necessary to learn a good comparator? We show here results with several alternatives to CONV5 features of AlexNet: 1) FC6 features of AlexNet, 2) CONV5 of AlexNet with random weights, 3) CONV5 of the network of Wang and Gupta [24] which we refer to as VideoNet. The results are shown in Fig. 6 . While the AlexNet CONV5 comparator provides best reconstructions, other networks preserve key image features as well.

**Sampling pre-images.** Given a feature vector $\mathbf{y}$, it would be interesting to not just generate a single reconstruction, but arbitrarily many samples from the distribution $p(\mathbf{I}|\mathbf{y})$. A straightforward approach would inject noise into the generator along with the features, so that the network could randomize its outputs. This does not yield the desired result, even if the discriminator is conditioned on the feature vector $\mathbf{y}$. Nothing in the loss function forces the generator to output multiple different reconstructions per feature vector. An underlying problem is that in the training data there is only one image per feature vector, i.e., a single sample per conditioning vector. We did not attack this problem in this paper, but we believe it is an interesting research direction.

### 4.2 Variational autoencoder

We also show an example application of our loss to generative modeling of images, demonstrating its superiority to the usual image space loss. A standard VAE consists of an encoder $Enc$ and a decoder $Dec$. The encoder maps an input sample $x$ to a distribution over latent variables $z \sim Enc(x) = q(z|x)$. $Dec$ maps from this latent space to a distribution over images $\tilde{x} \sim Dec(z) = p(x|z)$. The loss function is

$$\sum_i -\mathbb{E}_{q(z|x_i)} \log p(x_i|z) + D_{KL}(q(z|x_i)||p(z)), \qquad (7)$$

where $p(z)$ is a prior distribution of latent variables and $D_{KL}$ is the Kullback-Leibler divergence. The first term in Eq. 7 is a reconstruction error. If we assume that the decoder predicts a Gaussian distribution at each pixel, then it reduces to squared Euclidean error in the image space. The second term pulls the distribution of latent variables towards the prior. Both $q(z|x)$ and $p(z)$ are commonly

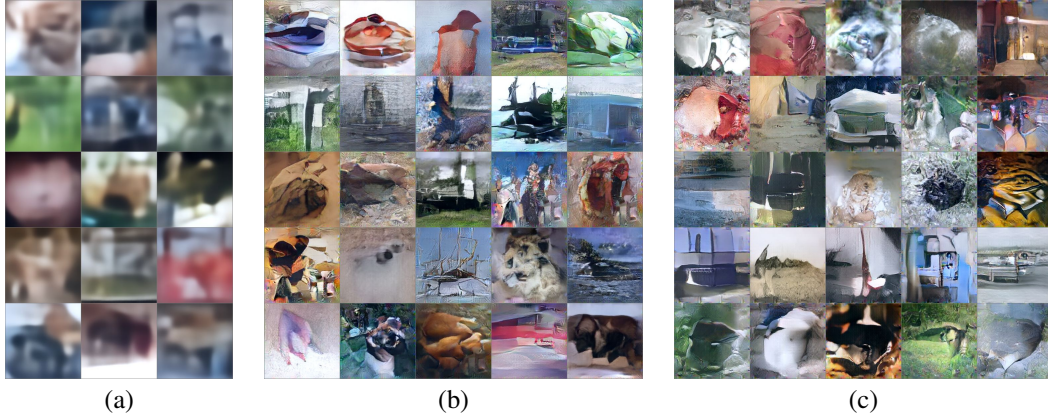

(a)                                    (b)                                    (c)

Figure 8: Samples from VAEs: **(a)** with the squared Euclidean loss, **(b)**, **(c)** with DeePSiM loss with AlexNet CONV5 and VideoNet CONV5 comparators, respectively.

assumed to be Gaussian, in which case the $KL$ divergence can be computed analytically. Please see Kingma and Welling [7] for details.

We use the proposed loss instead of the first term in Eq. 7 . This is similar to Larsen et al. [18], but the comparator need not be a part of the discriminator. Technically, there is little difference from training an "inversion" network. First, we allow the encoder weights to be adjusted. Second, instead of predicting a single latent vector $z$, we predict two vectors $\mu$ and $\sigma$ and sample $z = \mu + \sigma \odot \varepsilon$, where $\varepsilon$ is standard Gaussian (zero mean, unit variance) and $\odot$ is element-wise multiplication. Third, we add the KL divergence term to the loss:

$$\mathcal{L}_{KL} = \frac{1}{2} \sum_i \left( ||\mu_i||_2^2 + ||\sigma_i||_2^2 - \langle \log \sigma_i^2, \mathbf{1} \rangle \right). \tag{8}$$

We manually set the weight $\lambda_{KL}$ of the KL term in the overall loss (we found $\lambda_{KL} = 20$ to work well). Proper probabilistic derivation in presence of adversarial training is non-straightforward, and we leave it for future research.

We trained on $227 \times 227$ pixel crops of $256 \times 256$ pixel ILSVRC-2012 images. The encoder architecture is the same as AlexNet up to layer FC6, and the decoder architecture is same as in Table 1 . We initialized the encoder with AlexNet weights when using AlexNet as comparator, and at random when using VideoNet as comparator. We sampled from the model by sampling the latent variables from a standard Gaussian $z = \varepsilon$ and generating images from that with the decoder.

Samples generated with the usual SE loss, as well as two different comparators (AlexNet CONV5, VideoNet CONV5) are shown in Fig. 8 . While Euclidean loss leads to very blurry samples, our method yields images with realistic statistics. Global structure is lacking, but we believe this can be solved by combining the approach with a GAN. Interestingly, the samples trained with the VideoNet comparator and random initialization look qualitatively similar to the ones with AlexNet, showing that supervised training may not be necessary to yield a good loss function for generative model.

## 5   Conclusion

We proposed a class of loss functions applicable to image generation that are based on distances in feature spaces and adversarial training. Applying these to two tasks — feature inversion and random natural image generation — reveals that our loss is clearly superior to the typical loss in image space. In particular, it allows us to generate perceptually important details even from very low-dimensional image representations. Our experiments suggest that the proposed loss function can become a useful tool for generative modeling.

## Acknowledgements

We acknowledge funding by the ERC Starting Grant VideoLearn (279401).

## Footnotes

[1] An interesting alternative would be to explicitly analyze feature statistics, similar to Gatys et al. [2]. However, our preliminary experiments with this approach were not successful.

[2]This is unless the feature representation is specifically designed to map natural and non-natural images far apart, such as the one extracted from the discriminator of a GAN.

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
