[Supplementary Material]

# Supplementary material for "Generating Images with Perceptual Similarity Metrics based on Deep Networks"

**Alexey Dosovitskiy and Thomas Brox**
University of Freiburg
`{dosovits, brox}@cs.uni-freiburg.de`

Here we show additional technical details and results which could not be included into the paper because of space constraints.

**Discriminator architecture** is shown in Figure 1 . In our setup the job of the discriminator is to analyze the local statistics of images. Therefore, after five convolutional layers with occasional stride we perform global average pooling. The result is processed by two fully connected layers, followed by a 2-way softmax. We perform $50\%$ dropout after the global average pooling layer and the first fully connected layer.

**More inversion results** are shown in Figure 1 . The images are randomly selected from the ImageNet validation set.

**Nearest neighbors** of some validation images are shown in Figure 2 . We retrieved the nearest neighbors based on Euclidean distance in the pixel space and in the representations extracted from different layers of AlexNet. For comparison we also show the reconstructions produced by the network. Interestingly, for one of the images there is a near-duplicate in the training set. For other images the nearest neighbors are mainly very low quality when computed in the pixel space of low-level feature spaces, but very semantically meaningful when computed in high-level feature spaces. In most cases the reconstructions produced by the network better reflect characteristic properties of the images than the retrieved nearest neighbors. This is especially so for classes with large variability, such as dogs.

**Extended ablation study** of our loss function is shown in Figure 3 . Note how results without the image space loss are very similar to the results with the full loss, except for artifacts at the border and in some cases (second row) in the image.

**Position and color preservation** even in deep layers of AlexNet is illustrated in Figure 4 .

**Interpolation** between images in feature spaces is shown in Figures 5 and 6 . Note how interpolation between high level features lead to smooth transformations of objects shown in the images.

Samples from **variational autoencoders** with different losses are shown in Figure 7 . Fully unsupervised VAE with VideoNet [1] loss and random initialization of the encoder is in the bottom right. Samples from this model are qualitatively similar to others, showing that initialization with AlexNet is not necessary.

**Iterative encoding** of images to a feature representation and reconstructing back to the image space is illustrated in Figures 8 and 9 . As can be seen from Figure 9 , the network trained with loss in the image space does not preserve the features well, resulting in reconstructions quickly diverging from the original image.

| Type | conv | conv | conv | conv | conv | pool | fc | fc |
|---|---|---|---|---|---|---|---|---|
| InSize | 227 | 56 | 52 | 25 | 23 | 11 | — | — |
| OutCh | 32 | 64 | 128 | 256 | 256 | 256 | 512 | 2 |
| Kernel | 7 | 5 | 3 | 3 | 3 | 11 | — | — |
| Stride | 4 | 1 | 2 | 1 | 2 | — | — | — |

Table 1: Discriminator architecture.

Figure 1: Reconstructions of random validation images from higher layers of AlexNet. General characteristics of images are preserved very well. In some cases (simple objects, landscapes) reconstructions are nearly perfect even from FC8. In the leftmost column the network generates dog images from FC7 and FC8.

# References

[1] X. Wang and A. Gupta. Unsupervised learning of visual representations using videos. In *ICCV*, 2015.

Images

Nearest neighbors

Image

CONV1

CONV2

CONV3

CONV4

CONV5

FC6

FC7

FC8

Reconstructions

CONV5

FC6

FC7

FC8

Figure 2: Nearest neighbors in the image space and in feature spaces.

Figure 3: Reconstruction from FC6 with different components of the loss.

Figure 4: Position (first three columns) and color (last three columns) preservation.

Figure 5: Interpolation between cat images in the layer FC7.

Figure 6: Interpolation in feature spaces at different layers of AlexNet. **Topmost:** input images, **Top left:** CONV5, **Top right:** FC6, **Bottom left:** FC7, **Bottom right:** FC8.

Figure 7: Samples from VAE with our approach, with different comparators. **Top left:** AlexNet CONV5 comparator, **Top right:** AlexNet FC6 comparator, **Bottom left:** VideoNet CONV5 comparator, **Bottom right:** VideoNet CONV5 comparator with randomly initialized encoder.

Figure 8: Iterative re-encoding and reconstructions for different layers of AlexNet. Each row of each block corresponds to an iteration number: 1, 2, 4, 6, 8, 12, 16, 20. **Topmost:** input images, **Top left:** CONV5, **Top right:** FC6, **Bottom left:** FC7, **Bottom right:** FC8.

Figure 9: Iterative re-encoding and reconstructions with network trained to reconstruct from AlexNet FC6 layer with squared Euclidean loss in the image space. On top the input images are shown. Then each row corresponds to an iteration number: 1, 2, 4, 6, 8, 12, 16, 20.