[Reviews · NeurIPS 2016]

Reviewer 1

Summary

The paper proposes a class of loss functions to evaluate the quality of generated images. The loss is composed of three terms: (1) loss over image features extracted by a deep network, (2) loss over pixels, and (3) an adversarial loss. The methods leads to visually very impressive generations, significantly surpassing competing techniques.

Qualitative Assessment

The paper is exceptionally well written and the authors present very impressive results. The ideas in this paper are likely to be of use to the wider NIPS community. Line 119: `deconvolutional` -> `deconvolutional' Line 121: `bed of nails` -> `bed of nails'

Confidence in this Review

2-Confident (read it all; understood it all reasonably well)


Reviewer 2

Summary

The paper presents a method to generate images from high-dimensional vector representations using hybrid training objective called deepPSiM. The training objective is composed of three terms: 1) loss in the feature space: distance of feature embeddings between original image and generated image is small, 2) adversarial loss: generated image follows the data distribution in the input space via adversarial training, and 3) loss in image space: L2 pixel-wise distance between original image and generated image. The paper presents thorough experimental validation to show effectiveness of these components in the objective function in generating realistic images from high-dimensional representation.

Qualitative Assessment

I like the idea of adversarial training to generate less blurry and realistic looking images of objects from high-level representation. Although generative adversarial training has been extensively studied and compared with SE loss-based generative model (e.g., VAE), this work differentiates itself from previous work on generative models by generating realistic-looking images from very high-level representation of supervised CNN. It is quite surprising to me that even FC8 (softmax) layer representation has such a lot of information to invert back to image. I am wondering whether authors have observed any overfitting issue of adversarial training (Theis et al.). For example, have you compared with the nearest training example for inverted images of test example?

Confidence in this Review

3-Expert (read the paper in detail, know the area, quite certain of my opinion)


Reviewer 3

Summary

The manuscript 'Generating Images with Perceptual Similarity Metrics based on Deep Networks' proposes a distance measure between images that is meant to capture perceptual distances well. The distance measure consists of three terms: 1. A standard l2 distance in pixel space 2. A l2 distance in the feature space given by a late layer in a CNN (AlexNet here) 3. An adversarial loss that quantifies the probability with which a discriminator network thinks both images come from the same distribution. The manuscript presents two applications for that distance measure: 1. It shows that reconstructions of images from their compressed deep network representations look perceptually much closer to the original image when using the proposed distance measure as a loss function to train a generator network to invert the deep network representations. 2. It shows that images sampled from a VAE are of better perceptual quality when using the distance measure as the reconstruction loss term during training of the VAE. Furthermore it demonstrates how one can use the first application to smoothly interpolate between pairs of images by generating images from their linearly interpolated feature representations.

Qualitative Assessment

I think the most important contribution of the manuscript is to describe a method that substantially improves image reconstruction from compressed deep network representations (e.g. 4k dimensions vs. 227^2 * 3 = 150k dimensions), hence showing exciting new ways of images compression using deep networks. In that regard I would have liked an analysis of the compression rate for the reconstructions from the different feature spaces. In particular because the difference in quality between layer conv5 and fc6 doesn't seem too large, whereas there is a 10-fold reduction in dimensionality ((13 x 13 x 256 = 43k vs. to 4k) in the feature representation. One factor that should definitely be discussed in the paper is that it appears as if the adversarial prior enforces to use image data from the training set in the reconstruction. This is not necessarily a problem in terms of image compression, but is an important factor: a careful choice of training data might be important depending on what type of images one wants to compress. And the ability to reconstruct arbitrary images might be limited by the limit on available training data. The manuscript argues that the described method has potential applications in terms of reconstructing images from measured brain activity. This is indeed an exciting possible application. However, as there probably won't be enough neural data in the foreseeable future to train a generator from, this argument could have been made a lot stronger by showing that the learned generators are useful to reconstruct images from deep representations of new networks. E.g. one could train a linear predictor from a deep VGG representation to a deep AlexNet representation and show that one can still use the generator trained with AlexNet to reconstruct the image from the VGG representation. Minor comments/questions: - Was the comparator trained, it is mentioned that it could be trained in general but it doesn't say if it was in the described experiments. I assume it wasn't. - To what extend is random comparator different to just using the img + adv loss? There is only one example to compare and there it could be quite similar, meaning that the random representations don't contribute much. - The manuscript could mention that when training the generator for reconstruction that there is another prior enforced by the architecture of the generator in addition to the learned adversarial prior. This is important because the (convolutional) network architecture substantially limits the images that can be generated. In conclusion, I think that in general the manuscript is solid in its methods and arguments and demonstrates findings of importance to the field that will certainly influence future work.

Confidence in this Review

2-Confident (read it all; understood it all reasonably well)


Reviewer 4

Summary

This paper proposes a class of loss functions designed to more accurately invert features extracted by convolutional networks back into the image space. These combine three ideas: the standard pixel-wise squared loss, adversarial loss, and deep network feature comparators. In several qualitative results, the proposed model generates images that capture the sharpness and global structure in the original image, comparing favorably to the included baselines. However, many very similar image generation methods have been recently proposed (and published at ICML); it is not clear whether the proposed method offers a further performance benefit, and the novelty appears fairly low. Furthermore, I am skeptical of the proposed approach of treating image generation as a one-to-one mapping of embedding -> image, as is done in most of this paper. Ideally, a generative model should be able to condition on the embedding (which could even be a class label at the top-level of the encoder network), and draw diverse samples of images that could have produced it. If the task involves analyzing the invariance properties of the embedding, then looking at these diverse samples would provide a much better understanding of the embedding, by studying what things change or do not change in the image space. The authors propose a VAE version of their model in the end of the paper, but there are no results showing diverse samples conditioned on a single image encoding. Another way to get diverse samples would be to use adversarial training, which is already a part of the proposed model. What happens if the generator conditions also on a simple noise distribution when training with the adversarial loss? Does this enable the model to generate diverse samples? Overall, I appreciate the results of this paper, but I am not convinced it provides a huge amount of value beyond the previously-published works on image generation. I would not oppose it being published at NIPS, but I hope that the authors can address (1) the issue of learning one-to-many mappings from embeddings to image space, i.e. show that their model can draw diverse, realistic samples and (2) compare to the previous similar work appearing at ICML and ideally show how this work advances further.

Qualitative Assessment

For the VAE - it is not clear whether the samples have any diversity. What happens when you encode an image, and then draw multiple samples? Neither figure 8 nor figure 5 in the supplement make this clear. Several works referred to as concurrent, e.g. [18], should be considered significantly prior in my opinion. Another work that should be cited: “Augmenting Supervised Neural Networks with Unsupervised Objectives for Large-scale Image Classification” Yuting Zhang, Kibok Lee, Honglak Lee The 33rd International Conference on Machine Learning (ICML), 2016.

Confidence in this Review

3-Expert (read the paper in detail, know the area, quite certain of my opinion)


Reviewer 5

Summary

Measuring similarity between differences is difficult, and it is very clear that mean squared error is a bad choice of error, despite its popularity. For tasks where we seek to generate images, such as for unsupervised learning, such a loss function is crucial. This paper suggests that we define our loss as the sum of two terms: (1) the mean-squared error between features of the input and predicted images and (2) the negative log loss of a discriminator network that seeks to distinguish between samples from the input data and generated images. The rationale for this approach is that the first loss is insensitive to perceptually-unimportant variation, while the second term ensures that generated images have statistics typical of natural images.

Qualitative Assessment

Overall, the contribution of the paper is fairly incremental: both term (1) and (2) have been used before, just not their sum. These models are difficult to evaluate quantitatively. The only quantitative results in the paper are underwhelming (see 3 below). On the other hand, some of the results are quite compelling visually. A few more detailed comments: 1) The features from the comparator aren’t in a well-defined coordinate system. With ReLUs in particular, it would be easy to scale up one activation by alpha, and divide the weights by 1/alpha farther up in the network, and produce the exact same outputs. With this in mind, why is it reasonable to measure Euclidean distance in these feature spaces? Did you try anything like Mahalonobis distance, which would leverage the distribution of activations on some training data? 2) Table 3 presents the only quantitative results of the paper. Why did you omit results for your approach for everything but Fc6? These seem like important results for the reviewers to be able to evaluate the effectiveness of your method. 3) While the results in Table 4, etc. are impressive, I’m unconvinced by your argument for why ‘inverting' neural network features is an important problem. Besides being an exploratory tool to help us understand the signal being captured by an existing network, why is this useful? I’m unconvinced that this paper provides any progress towards your goal of applying the method to real brain recordings, especially since you rely on differentiability of the comparator. 4) Most contemporary papers on generative image models have attempted to convince readers that their model didn’t just ‘memorize’ the training data. With such high-capacity models, this is a real risk. How should we be confident that you didn’t achieve memorization in your VAE experiments?

Confidence in this Review

3-Expert (read the paper in detail, know the area, quite certain of my opinion)


Reviewer 6

Summary

Authors propose to use adversarial networks as a loss function to measure the quality of the image generated by inverting feature representations of neural nets. The proposed method brings together a set of existing modules and the results are good.

Qualitative Assessment

The paper is easy to read and understand. The proposed loss is well motivated and makes sense. Even though the technical novelty is limited, the results are good. If the authors are willing to include random samples of reconstruction, I think the paper is good enough to be a poster.

Confidence in this Review

3-Expert (read the paper in detail, know the area, quite certain of my opinion)